# Immune Stroma in Lung Cancer and Idiopathic Pulmonary Fibrosis: A Common Biologic Landscape?

**DOI:** 10.3390/ijms22062882

**Published:** 2021-03-12

**Authors:** Sara Lettieri, Tiberio Oggionni, Andrea Lancia, Chandra Bortolotto, Giulia Maria Stella

**Affiliations:** 1Department of Medical Sciences and Infective Diseases, Unit of Respiratory Diseases, IRCCS Policlinico San Matteo Foundation and University of Pavia Medical School, Viale Camillo Golgi, 19, 27100 Pavia, Italy; sara.lettieri01@universitadipavia.it (S.L.); t.oggionni@smatteo.pv.it (T.O.); 2Department of Medical Sciences and Infective Diseases, Unit of Radiation Therapy, Fondazione IRCCS Policlinico San Matteo, Viale Camillo Golgi, 19, 27100 Pavia, Italy; a.lancia@smatteo.pv.it; 3Department of Intensive Medicine, Unit of Radiology, IRCCS Policlinico San Matteo Foundation and University of Pavia Medical School, Viale Camillo Golgi, 19, 27100 Pavia, Italy; c.bortolotto@smatteo.pv.it

**Keywords:** fibrosis, cancer, immune system, microenvironment, genetics, targeted therapy

## Abstract

Idiopathic pulmonary fibrosis (IPF) identifies a specific entity characterized by chronic, progressive fibrosing interstitial pneumonia of unknown cause, still lacking effective therapies. Growing evidence suggests that the biologic processes occurring in IPF recall those which orchestrate cancer onset and progression and these findings have already been exploited for therapeutic purposes. Notably, the incidence of lung cancer in patients already affected by IPF is significantly higher than expected. Recent advances in the knowledge of the cancer immune microenvironment have allowed a paradigm shift in cancer therapy. From this perspective, recent experimental reports suggest a rationale for immune checkpoint inhibition in IPF. Here, we recapitulate the most recent knowledge on lung cancer immune stroma and how it can be translated into the IPF context, with both diagnostic and therapeutic implications.

## 1. Introduction

Idiopathic pulmonary fibrosis (IPF) is defined as a chronic fibrosing interstitial lung disease of unknown etiology that occurs in older adults and is characterized by the histopathological pattern of usual interstitial pneumonia (UIP) [1]. Histologic criteria required to confirm UIP diagnosis are represented in Figure 1. With an estimated incidence of 10.7 cases per 100,000 per year for males and 7.4 cases per 100,000 per year for females, and an estimated prevalence of 20/100,000 for males and 13/100,000 for females [2,3], IPF is the most representative of the interstitial lung diseases (ILDs). Despite the progress in the knowledge of pathogenesis and the increasing number of clinical trials dedicated to this disease, prognosis of IPF is still poor, with a median survival of 3-5 years after diagnosis, similar to or worse than that of several oncologic diseases [4,5]. The clinical course is highly variable and terminal respiratory failure due to disease progression is the most common cause of death. It should be underlined that treatment with pirfenidone and nintedanib, currently approved in clinical use, has significantly improved survival and reduced disease progression, although no significant effect has been demonstrated on quality of life [6].

Moreover, in many patients, rate of decline is negatively influenced by episodes of acute respiratory worsening that may be associated with high mortality. These events could be secondary to infections, pulmonary embolism, and pneumothorax or hearth failure [7]. If the underlying cause is not defined, these events are termed “acute exacerbations” or “accelerated phase” of IPF [8]. Extrapulmonary and pulmonary comorbidities, such as lung cancer, can associate to IPF, further varying prognosis.

### IPF: Pathobiologic Traits Recalling Malignant Proliferation

As well as sharing risk factors such as smoking of cigarettes, older age, and male sex, IPF is an independent cancer risk factor and may be regarded as a precancerous lesion [9,10,11,12,13]. Prevalence of cancer in the IPF population ranges from 2.8 to 48% [14], increases with each year after IPF diagnosis, and is associated with a reduced mean survival time (1.6–1.7 years) [15,16]. The most frequent histotype is squamous cell tumor (SCC) [17,18], followed by adenocarcinoma (ADC); small-cell-lung cancer (SCLC) is not infrequent [17,18,19,20]. Lung cancers are found more frequently in peripheral zones of lungs, where fibrosis and honeycombing predominate [21]. These findings have given rise to increasing interest in understanding common traits between IPF and cancer.

According to the currently accepted pathogenic hypothesis, environmental and/or occupational factors such as smoking of cigarettes, pollution, dust exposure, infections, particularly viral ones (EBV, CMV, HCV, HHV-8), and repeated microaspiration of digestive juices in gastroesophageal reflux disease (GERD), acting in genetically susceptible individuals, cause repetitive injury to the aged alveolar epithelium, which triggers aberrant wound healing repair processes and initiates the accumulation of extracellular matrix (ECM) deposited by activated myofibroblasts. The production of these pro-fibrotic cells is due to the activation of epithelial–mesenchymal transition (EMT), a dynamic process where polarized epithelial cells undergo molecular and phenotypic changes, allowing them to gain a mesenchymal and undifferentiated potential. This plastic change implies that cells acquire, on one hand, an increased ability to produce ECM components whereas, on the other, they become more motile, with higher invasive properties [16,18]. The result of a complex system of interactions of genetic, epigenetic, and environmental factors is progressive lung remodeling, architectural distortion, and fibrosis. The pathobiologic events of IPF strictly resemble those occurring in malignant transformation. IPF and cancer share features in terms of genetic alterations, abnormal expression of microRNAs, aberrant activated pathways, altered crosstalk communication, responses to proliferative and inhibitory signals, resistance to apoptosis, and myofibroblast origin and behavior [21]. The recognition of common pathogenic pathways between cancer and IPF has been stimulating the development of new therapeutic strategies. Receptor tyrosine kinases (RTKs) are key regulatory signaling proteins governing cancer cell growth and metastasis. More recently, their aberrant expression has been associated with wound healing and fibrogenesis. The antifibrotic activity of receptor tyrosine kinase inhibitors (RTKIs) has been demonstrated in vitro and in vivo, suggesting a possible role of RTKIs as a novel therapeutic approach for the management of IPF [22,23]. This partially became a reality in 2014, when the Food and Drug Administration (FDA) approved the multi-kinase inhibitor nintedanib, originally designed as an anti-angiogenic agent against cancer, for the treatment of IPF [24]. The aim of this review is to explore the common pathogenetic pathways and the common biologic background between IPF and cancer, with a focus on the role of the surrounding microenvironment, in order to address the urgent need for a better comprehension of the molecular mechanisms of IPF and to make them potentially druggable by specific molecules, similarly to targeted cancer therapy.

## 2. Cancer and IPF: Genetic, Epigenetic, Signaling Links and Differential Traits

Some of the dynamic processes which characterize cancer [14] appear to be disrupted in both IPF and cancer. Tissue homeostasis is guaranteed by cell-to-cell communication through gap junctions, membrane channels formed by connexines (Cxs), essential for electro-metabolic synapse between adjacent cells. Connexin 43 (Cx43) is the most represented connexin on fibroblasts and the alveolar epithelium; it is involved in intercellular crosstalk and, in cooperation with caveolin 1, protect cells from entering the EMT process [25,26]. Inappropriate downregulation of Cx43 is associated with enhanced tissue repair, which can result in uncontrolled fibroblast proliferation and ultimately in IPF [27]. Similarly, in lung cancer cells, low or absent expression of Cx43 results in a reduction in cell-to-cell communication, loss of contact-inhibition control, and aberrant proliferation, events which characteristically mark this disease [28]. Caveolin 1 (Cav 1) has been suggested to act as an antifibrotic molecule [29]; however, controversial data have been published [30]. It is conceivable that due to the varied roles played by Cav1, it might function in cancer both as an inhibitor and a promoter of growth signaling according to the disease’s temporal and spatial stage [31].

### 2.1. Genetic Alterations

#### 2.1.1. Oncogenes

Among cancer genes, receptor tyrosine kinases (RTKs) are known to play a driving oncogenic role [32]. They are tightly regulated, and their aberrant activation, through the occurrence of activating genetic lesions (somatic point mutations, increased gene copy number, translocations), triggers oncogenesis, leading to the constitutive activation of a great number of downstream signaling pathways. The activated kinases represent particularly good drug targets, exploiting the phenomenon of “oncogenic addiction” (the dependence of tumor cells on a single oncogenic activity for their survival and proliferation), that of “oncogenic shock” (the apoptotic death of neoplastic clone due to loss of pro-survival signals and predominance of pro-apoptotic signals following oncoprotein inactivation), or both [33]. TK activation has also been shown in IPF, as demonstrated, for instance, by the marked immunohistochemical expression of phosphorylated mTOR, phosphorylated ERM, PTEN, MET, and, in approximately 15% of cases, EGFR in UIP samples [34]. However, the mechanisms leading to overexpression of RTKs in IPF may be different from those of cancer, and TK protein expression may not necessarily imply increased activity. Moreover, it is unlikely that IPF fibroblasts may be addicted to TK activation for growth and proliferation, so oncogenic shock cannot be exploited for therapeutic purposes in IPF. Rather than being a driving mechanism conferring a clonal growth advantage, TK activation in IPF indicates an expedient exploited to promote inappropriate fibrotic proliferation. Coherently, the therapeutic blockade of RTKs in IPF represents an approach to hamper progression in the absence of clonal evolution phenomena [28]. Among proto-oncogenes, the most relevant example regards epidermal growth factor receptor (EGFR), a transmembrane receptor tyrosine kinase belonging to the erbB family [35,36]. EGFR is normally expressed in several tissues but is overexpressed or mutated in 43–89% of NSCLC (especially adenocarcinoma histology, history of never smoking, women of Asian ethnicity) [37], where it is associated with reduced survival [38,39]. However, the introduction of reversible EGFR TKIs, gefitinib or erlotinib, or the second-generation, irreversible TKI, afatinib, has improved the prognosis of these patients, although acquired resistance almost inevitably develops, on average, one year after the initiation of TKI therapy. Increased expression of EGFR, in both protein levels and mRNA, has been documented also in IPF, COP, and fibrotic NSIP, primarily in the hyperplastic alveolar epithelium surrounding areas of fibrosis, suggesting a possible role for this receptor in the aberrant reepithelization that characterizes lung inflammation and fibrosis. We and others reported that the EGFR pathway is phosphorylated in IPF areas and that the activation does not rely on the occurrence of genetic lesions, which are rarely detected, but rather on an EGFR-dependent paracrine loop between epithelial and fibroblast cells, resulting in disproportionate collagen production and deposition [40,41]. Nevertheless, the pathogenic role of the infrequently reported activating mutation remains to be clarified since it could potentially represent an early marker of malignant transformation inside UIP parenchyma [42]. Interestingly, expression of EGFR is more pronounced in fibrotic forms of ILDs compared with inflammatory forms of ILDs and in more severe forms of fibrosis, suggesting a harmful role of EGFR in the process of fibrogenesis [43]. Overall, these observations allow us to exclude a context of EGFR addiction of fibroblasts and this point limits the use of anti-EGFR agents developed for cancer therapy. In vivo, EGFR pharmacologic inhibition has shown controversial effects [40,44]. Moreover, despite the proven efficacy of TKIs in cancer, their role in IPF remains debated due to their potential pulmonary toxicity and fibrosis induction capacity [44,45]. Finally, other elements, such as cigarette smoking, could influence responses to TKIs. For instance, several studies have reported that cigarette smoking is an independent negative predictive factor, since it not only activates EGFR but also stabilizes the receptor against degradation. Thus, the exact meaning of *EGFR-*activating mutations in IPF remains elusive.

#### 2.1.2. Tumor Suppressor Genes

Among tumor suppressor genes, it is well known that p53 mutations occur during the early phases of lung carcinogenesis [46,47]. Frequent *p53* tumor suppressor gene alterations have also been detected in atypical epithelial lesions in the peripheral areas of fibrotic lungs, where neoplastic lesions more frequently develop, suggesting a possible role of p53 in the early stages of carcinogenesis in patients with IPF [48,49]. Together with p53, also p21 and p16, overexpressed in lung cancer, are highly detectable in epithelial cells of remodeled areas of UIP, where they are markers of cellular senescence and may contribute to the development of cancer through the induction of a senescence-associated secretory phenotype (SASP) [50]. However, loss of heterozygosity, homozygous deletions, and abnormal expression of the human fragile histidine triad (*FHIT*) tumor suppressor gene have been described in NSCLC, particularly in SCCs and in smokers, and in metaplastic areas and bronchiolar epithelia of IPF patients [51].

### 2.2. Microsatellites and Telomeres

Cancer is also marked by several genetic alterations, namely micro-satellite instability and loss of gene heterozygosity, that usually correlate with a high rate of mutations and with repair of DNA damages. In approximately 50% of IPF cases, microsatellite instability and loss of heterozygosity, targeting genes such as *MYCL1*, *FHIT*, *SPARC*, *p16Ink4,* and *TP53* have also been reported, suggesting that, autonomously, IPF exhibits a high mutational background which can ultimately lead to cell cycle and apoptosis deregulation [10]. Typically regarded as “aging diseases”, both pulmonary fibrosis and cancer share telomerase dysfunction. The telomerase is a reverse transcriptase enzyme that adds TTAGGG repeats to the 3’ chromosomal ends during mitosis, preventing premature telomere shortening, and is composed of three subunits, the telomerase reverse transcriptase (h-TERT), displaying catalytic activity, the RNA subunit (h-TERC), and dyskerin [52,53,54,55]. Telomerase subunits are not expressed in most somatic cells, where they act as a “mitotic clock”, while they are highly expressed in germ cells and in neoplastic immortalized cells [56,57,58]. It is known that inherited mutations of TERC and TERT genes are associated with 8–20% of familial IPF and with 1% of sporadic IPF [55,59,60]. Moreover, approximately 37% of familial and 25% of sporadic cases of IPF without TERT and TERC mutations are associated with lengths of telomeres lower than the 10th percentile compared to the general population [61]; impairment of telomerase expression in IPF reveals a differential function played by the enzyme in fibrogenesis vs. carcinogenesis. However, as in cancer, telomerase inhibition has been exploited for therapeutic purposes [62,63].

### 2.3. Epigenetic Determinants

It is well established that methylation of suppressor genes or hypomethylation of oncogenes, driven by genetic and environmental factors such as aging, cigarette smoking, exposition to pneumotoxic agents, or diet, are involved in the initiation and progression of cancer. These epigenetic changes have been similarly observed in IPF. For instance, reduced expression of the Thy-1 glycoprotein, due to the hypermethylation of the promoter region of the gene coding for this glycoprotein, is associated with the increased motile capacity of malignant cells [64]. In IPF, the loss of its expression induces the de-differentiation of fibroblasts into myofibroblasts and the generation of fibroblast foci [65,66,67]. Intriguingly, the pharmacological blockage of Thy-1 methylation could be exploited for therapeutic purposes. However, reduced expression of Smad4 due to hypermethylation of the promoter region of the gene, crucial in pulmonary carcinogenesis, has been implicated in the progression of IPF through the overexpression of TGF-ß [68]. Conversely, promotion of the hypermethylation of the O-6-methylguanine DNA methyltransferase (MGMT) gene is a distinctive feature of lung cancer [69], while in IPF fibroblasts, MGMT is hypomethylated [70], demonstrating, however, the existence of disease-specific methylation patterns. Growing evidence has demonstrated that dysregulation of miRNAs contributes to the development of cancer, promoting angiogenesis, tissue invasion and metastasis, sustaining proliferative signaling and escaping growth suppressors, through various mechanisms, such as amplification or deletion of miRNA genes, abnormal transcriptions, altered epigenetic changes, and defects in the miRNA biogenesis [71]. These small non-coding RNA molecules have been studied as actionable targets with prognostic and predictive implications. Around 10% of miRNAs are found to be dysregulated in IPF, affecting processes such as EMT or apoptosis regulation. Some of them are upregulated (e.g., miR21, miR155) while others are downregulated (e.g., let-7, miR-29, miR-30, miR-200, miR-423, miR-210, and miR-185) [72,73]. In particular, miR21 is upregulated in both IPF and lung cancer: it shows profibrotic activity, promoting TGF beta signaling through downregulation of Smad7, a negative regulator of TGF-beta, in human IPF and in murine bleomycin-induced fibrosis [74,75], and acts as an oncogene in NSCLC, where it represents an independent negative prognostic factor [76]. In experimental models, the administration of antisense miR21 probes attenuated the severity of fibrosis in mice, opening the way to novel therapeutic approaches [75]. Similarly, miR-155, an onco-miR overexpressed in IPF, promotes collagen synthesis through two profibrotic pairs, the Wnt/β-catenin and Akt signaling pathways. The evidence that miR-155 knockout mice are resistant to bleomycin-induced skin fibrosis should be exploited for the development of miR-155 inhibitors [77]. Few data are instead available regarding the role of miRNAs in modulating the IPF microenvironment: let-7, miR-30, miR-29 family, miR-335, and miR-338 (collectively defined as antifibrotic microRNAs) inhibit the ECM remodeling of the extracellular matrix and growth factors [78,79], while the hsa-miR-486-3p is known to be associated with nintedanib treatment but can also reduce angiogenesis; on the contrary, overexpression of the miR-192 analogue induces ECM production [80,81]. It should be underlined that nintedanib has been previously used as a second-line therapy in combination with docetaxel for advanced lung cancer. A recent bioinformatic analysis evaluating oncogenic TK expression in IPF pointed out five microRNAs which act by modulating the VEGF-A signaling pathway and epithelial to mesenchymal transition mechanisms in IPF [82]. This evidence strongly supports the rationale of clinical use of nintedaninb based on a cancer-related milieu which plays a central role in IPF [14,83,84].

### 2.4. Biologic Divergences

Despite the biological similarities between IPF and cancer, these diseases remain distinct in critical aspects: first, IPF is a lung-specific disease, with no ability of distant cell scattering, in contrast with the acquired ability of the neoplastic cells to distant dissemination and spreading [14]; IPF is bilateral, whereas, usually, cancer originates as a unilateral lesion and, after having metastasized, invades other tissues and organs, except for cases of multiple synchronous lesions. IPF is a heterogeneous disease in both the temporal and spatial interval of lesions. Indeed, the distribution of key histologic lesions, namely the fibroblastic foci (FF), follows a centripetal direction and is in contrast with neighboring areas of healthy parenchyma. Moreover, myofibroblasts in fibroblastic foci are polyclonal, whereas neoplastic cells are monoclonal [49]; in this regard, cancer is a genic disease, which evolves thorough a dynamic process of clonal expansion and selection of advantageous somatic driver lesions. IPF is characterized by an excess of persistently activated myofibroblasts derived from fibroblasts and from epithelial cells type II through the process of epithelial–mesenchymal transition (EMT) in a process involving TGF-ß [85]. The activated myofibroblasts are stellate or spindle-shaped cells expressing both epithelial and mesenchymal markers, such as α-smooth muscle actin (α-SMA), vimentin, and type I collagen [86]. Myofibroblasts sustain their growth by secreting in an autocrine way the fibrogenic cytokines TGF-ß and reducing the production of the anti-fibrotic prostaglandin E_2_ (PGE_2_). Similarly to IPF, cancer-associated myofibroblasts produce growth factors such as TGF-ß and metalloproteinases, which degrade the extracellular matrix, facilitating the invasion of cancer cells through the surrounding tissues. Interestingly, molecules typically associated with cancer invasiveness and metastasis, such as laminin, heath shock protein (HSP)27, and fascin, are also expressed in the epithelial cells surrounding fibroblast foci, where they are involved in the migration and invasion of basal cells adjacent to fibroblasts [87,88,89]. With respect to the activation of epithelial-to-mesenchymal transition, which characterizes both cancer and fibrosis, main efforts should be directed toward the proto-oncogene MET [90]. We and others already detected the expression of phospho-MET in both myofibroblasts and epithelial cells in IPF [91]. In neoplastic settings, therapeutic targeting of MET can occur as a front-line intervention (for the small fraction of addicted tumors featuring MET activation related to genetic lesions, namely mutation, amplification, translocation) and as an adjuvant approach in those neoplastic lesions which take advantage of MET activation by transcriptional upregulation for progression and dissemination. Critical IPF features recall MET-related cancer behaviors, such as branching morphogenesis and activation of clotting cascade. However, IPF is a lung-specific disease, in the absence of distant cell scattering and clonal selection and evolution.

## 3. Stemness-Like Networks

A great number of signal cascades are shared by carcinogenesis and fibrogenesis. The WNT/beta-catenin pathway regulates tissue plasticity and remodeling through the expression of several molecules, such as matrilysin, laminin, and cyclin-D1, and is involved in a strict crosstalk with TGF-ß. WNT/beta-catenin is overexpressed in several human carcinomas and desmoid tumors [13,67]; in IPF, hyperexpression of this signaling pathway has been localized in different pathologic areas, namely in bronchiolar proliferative lesions, damaged alveolar structures, and fibroblast foci [92]; dysregulated WNT signaling activation may lead epithelial cells to produce TGF-ß, to induce EMT, and to produce extracellular matrix (ECM) components. Interestingly, in murine models of bleomycin-induced fibrosis, the administration of pharmacological inhibitors of WNT signaling resulted in the amelioration of fibrosis and improved survival [93]. The sonic Hedgehog (SHH) is an embryological pathway crucial during lung development: it is expressed by the distal epithelium, where it serves as a ligand for receptors on the surrounding mesenchymal cells and drives the normal branching of the lung. SHH is not expressed in the adult lung in normal conditions but it has been found to be aberrantly reactivated in IPF, mainly in the epithelial cells surrounding honeycombing cysts, where it promotes the proliferation, migration, and survival of fibroblasts, collagen production, and fibronectin expression [94,95]. This pathway is also activated by cancer stem cells in the early stages of carcinogenesis and acts in a paracrine way on other neoplastic cells, leading to tumor growth and spread through EMT. In cancer cells, SHH activation in canonical and non-canonical ways is associated with drug resistance [96]. Notch is another embryological pathway normally active during lung development in both epithelial and mesenchymal compartments; it is involved in cell differentiation and migration, recruitment and determination of vascular smooth muscle cells, EMT, and expansion of Clara cells [97]. In adult lungs, reactivation of Notch signaling is associated with abnormal myofibroblasts and EMT [98]. Similarly, aberrant Notch expression has been reported in several human tumors. The exact role of Notch in tumorigenesis depends on the cellular context and microenvironment, with several isoforms exhibiting different functions (e.g., Notch1 acts as an oncogene whereas Notch2 is a tumor suppressor), complicating the development of Notch-targeted therapies [99]. Finally, the phosphoinositide 3-kinase (PI3K)/protein kinase B (AKT)/mammalian target of rapamycin (mTOR) signaling, one of the most deregulated pathways in NSCLC, regulates cell proliferation and resistance to apoptosis [11]. Recently, the discovery that overexpression of PI3K class I isoform p110ɣ occurs in both IPF lung homogenates and ex vivo fibroblast cell lines and that p110ɣ pharmacological inhibition or gene silencing can inhibit cell proliferation and α-smooth muscle actin expression in IPF fibroblasts have been opening the way to novel pharmacological approaches [100]. In this sense, on the basis that administration of p110γ inhibitors has been shown to significantly prevent bleomycin-induced pulmonary fibrosis in rats [101], several experimental studies have been evaluating the efficacy of mTOR/PI3K inhibitors for the treatment of IPF [102].

## 4. Common Available Biomarkers

The biologic relation which subsists between cancer and IPF gives a strong rationale for the identification of available biomarker signatures able to stratify patients based on the risk of disease progression and poor outcome. The perspective observational PROFILE study allowed the generation of a panel of serum proteins secreted from the metaplastic epithelium, aberrantly activated in the fibrogenic process that is most associated with IPF progression if compared to controls. They were surfactant protein D (SP-D), α2 macroglobulin, matrix metalloproteinase 7 (MMP7), T-cadherin, AXL receptor tyrosine-kinase, and MMP9. Notably, high levels of the CA19-9 protein were significantly associated with progressive disease, whereas high levels of other proteins such as CA-125, macrophage migration inhibitory factor (MIF), carcinoembryonic antigen (CEA), free prostate-specific antigen (PSAf), and MMP7 were associated with a significant increase in overall mortality [103]. Among those serum proteins, oncomarkers are routinely measured in biochemistry laboratories. With respect to their predictive role, it has been reported that their combination in a specific signature (high levels of circulating CEA, Ca15.3, Ca19.9, Ca125, and KL-6) is clearly associated with IPF [104]. Importantly, concomitant histologic studies demonstrated that the Ca19.9 and Ca125 markers are derived from epithelial damage [105]. By stratifying patient in terms of disease progression rate, Ca 19.9 level emerges as being inversely correlated to lung function decline, mainly in those patients who rapidly progress. Increased Ca 19.9 levels are associated with increased levels in the end-stage IPF, irrespective of treatment setting [106]. Indeed, fewer data are available on the potential role of circulating oncomarker levels in predicting response to therapy. The KL-6 levels have been reported to correlate with responses to nintedanib in IPF patients, since they decrease during treatment and, in the case of KL-6 levels, they declined during treatment and were associated with pulmonary function test performance [107]. Similarly, changes in CA 15-3 levels predict variation in functional tests in IPF on antifibrotic treatment and, coherently, higher levels are associated with worse outcomes [108]. This observation points out the role of the Ca 15.3 marker as a powerful and easily measurable follow-up biomarker in IPF. Overall, the described serum biochemical signatures support the evidence of a common biologic landscape between IPF and (lung) cancer, on one hand, whereas, on the other, they represent easy, non-invasive, and non-expensive reliable IPF markers, potentially powerful to predict responses to drugs and disease progression (and eventually to rapidly identify those patients who will require strict follow-up and will be earlier referred for lung transplantation).

## 5. Is There a Role for Immune Checkpoints in IPF?

In lung proliferative disease, either neoplastic or not (e.g., pulmonary fibrosis), it is not only the proliferating cell itself at the center of the pathologic mechanism but also a variety of resident and infiltrating cells, growth factors, and extracellular matrix proteins that overall define the microenvironment. Interactions of proliferating cells with their environment deeply influence the disease development and natural history and ultimately determine whether the disease progression can be controlled or not. Most of the data regarding IPF microenvironment composition and its dynamic evolution have been obtained from neoplastic samples featuring usual interstitial as concomitant disease. The tumor-surrounding stroma is known to intervene in response to anticancer drugs and represents an independent powerful target, as demonstrated by the paradigm of immune checkpoint inhibition. Notably, an immunosuppressive tumor microenvironment is characteristically associated with the development of UIP associated with lung cancer [109]. Among tumor-infiltrating lymphocytes (TILs), the number of CD8+ is significantly lower in those cases carrying IPF, whereas the CD8+/Foxp3+ T cell ratio was higher than that in the non-IPF cancer population [110]. The D8/FOXP3 T cell ratio is reported to be lower (P = 0.062) in cancer-associated areas that are known to be associated with cancer progression and represents a pro-fibrotic stroma element [111]. It has been demonstrated in many cancer types that the epithelial-to-mesenchymal transition (EMT) status is related to programmed death-1 ligand 1 (PD-L1) expression and is implicated in immune evasion [112,113,114]. Among regulators of both processes, in cancer, the transcription factor NFκB and INF-γ are known to play a key role. NFκB directly activates the *PD-L1* gene promoter and regulates PD-L1 at the post-transcriptional level through indirect pathways [115]. On the other hand, the NFκB pathway is required for EMT persistence, which is essential for neoplastic spreading [116,117]. Similarly, in vitro experiments have shown that tumor cells (pancreatic dutcal adenocarcinoma) promote PD-L1 expression and EMT in the presence of IFN-γ [118]. Although documented in malignant settings, these observations are of relevant interest as potentially translatable in fibrosing diseases such as IPF. Indeed, NF-κB induces the expression of proinflammatory cytokines which promote profibrotic inflammatory milieu [119,120,121]. Interferon-γ is a proinflammatory cytokine and is a key activator of macrophages. In contrast, it can block fibroblast proliferation and collagen production [122,123]. For these biologic effects, INF-γ has been exploited to treat not only lung fibrosis but also liver and renal fibrosis [124]. The INSPIRE study showed that INF-γ b given subcutaneously did not impact the outcomes of IPF patients [125]. However, IFN-γ seems to show a synergistic effect with pirfenidone [124].

Immune checkpoints (ICs) are defined as molecules on the surfaces of cells that can send inhibitory stimuli to attenuate immune responses. Tumor cells express checkpoint proteins on their surfaces to evade host immune responses. Targeted inhibition towards these receptors enhances the T-cell response towards the tumor [126,127]. Cytotoxic T-lymphocyte antigen 4 (CTLA-4), programmed cell death-1 PD-1, and programmed cell death-1 ligand (PD-L1) are key negative regulators of anti-tumor T-cell reactivity. The development of immune checkpoint inhibitors has revolutionized the treatment of a variety of cancers. An exhaustive review of immune checkpoint inhibitions in cancer is beyond the scope of this work. In the neoplastic setting, it has been demonstrated that immune checkpoints are expressed also in stromal cells [128,129,130] and that their selective modulation might impact the responses of tumor cells to IC blockade [131]. Overall, immune checkpoints behave as regulators of the immune system and self-tolerance and their expression has been already reported in lung-fibrosing areas (Figure 2), even though no clinical trial has been till now registered regarding the use of IC inhibitors in IPF (search of: immune checkpoint inhibitor | fibrosis—List Results—ClinicalTrials.gov). Experimental findings showed that the activation of PD-L1 in IPF fibroblasts promoted invasion in vitro and pulmonary fibrosis in vivo [132]. Preliminary data on small patient cohorts reported that high levels of serum PD-1/PD-L1 could be detected in patients with IPF and should potentially become non-invasive prognostic or/and predictive disease biomarkers [133]. In IPF, PD-1 expression has been detected in T lymphocytes both in blood and lung parenchyma [134]. It has been shown that PD-1 expression on CD4+ T cells can induce STAT3 upregulation and subsequent overexpression of IL-17a and TGF-β. Moreover, PD-1 seems to inhibit the differentiation of CD4+ T cells into Treg cells and to increased collagen-1 production [135]. Coherently to these preliminary findings, administration in animals of immune checkpoint inhibitors attenuates fibrosis [136]. Fewer elements are currently available regarding the role of CTLA-4, even though it has been reported to be overexpressed in IPF as compared to hypersensitivity pneumonitis [137]. The affinity of CTLA4 towards its B7 family ligands, CD80 and CD86, is stronger than the affinity of their cognate stimulatory coreceptor CD28. A loss of CD28 transcript expression, in association with the recruitment of proinflammatory cytokines, has been reported in CD3+ T cells isolated from IPF lung explants vs. healthy lungs. Notably, the IPF-derived T cells enriched with CD28^null^ elements can activate lung remodeling in mice. The CD28^null^ T cells expressed high levels of both CTLA4 and PD-1 molecules, and in in vivo models, the use of anti-CTLA-4 and PD1 agents seemed to impact on this process [138].

The importance of these signaling pathways, and of the common biologic landscape that they reveal, is not limited to treatment (e.g., anti-PD1 agents) but also to the diagnosis and specifically to IPF imaging. Positron emission tomography (PET) has proven to offer some interesting results in patients with IPF since FDG lung uptake functions as a marker of disease severity and outcome [139]. To enhance PET’s ability to become a biomarker of disease presence or progression, some new tracers have been proposed. These tracers are tightly linked to the pathways and biologic landscape explored throughout this review, such as integrin αvβ6 cystine knot or type I collagen-targeted PET probes [140,141]. Other imaging modalities may exploit the common morpho-structural characteristics of IPF and tumors, such as the associated fibrosis. The use of lung magnetic resonance (MR) and elastic registration to assess pulmonary fibrosis needs to be further explored. Functional imaging biomarkers can play an important role in monitoring IPF progression and response to therapies [142]. Additionally, in MR, the ratio of the signal from the ^129^Xe uptake in red blood cells (RBCs) to the signal of the gas uptake in the alveoli barrier was significantly reduced in patients with IPF, as a direct consequence of early interstitial thickening [143].

## 6. Conclusions

IPF is a devasting disease still lacking effective therapies. Initially, pathogenetic theories, considering IPF as the result of chronic inflammation further inducing fibrosis, led to the use of anti-inflammatory and immunosuppressive agents, which were revealed to be largely ineffective. Along with the new pathogenetic concept of IPF as an aging pathology due to chronic injury of dysfunctional epithelial cells, the therapeutic landscape also changed, focusing on targeting fibrogenesis. Taking advantage of progress in cancer biology and genome projects, the role of TK activation in IPF is now well established. However, the histologic heterogeneity and the consequent absence of genetic addiction phenomena clearly defines the context of aberrant fibroproliferation vs. clonal malignant expansion. In a similar fashion, the more recent advances obtained by immunotherapy in cancer have opened the way to a paradigm shift in IPF. This should represent an exceptional opportunity to improve the understanding of the biology of this orphan disease as well as to validate novel biomarkers and to design more effective therapeutic strategies. From this perspective, the novel advances regarding the critical role played by the microenvironment in tumor progression, dissemination, and resistance to therapy, and as a potent therapeutic target, can be translated to the afinalistic fibrosing setting, as represented by IPF, to identify and validate powerful disease biomarkers and to exploit the potential of immune checkpoint inhibition in blocking disease evolution.

## Figures and Tables

**Figure 1 ijms-22-02882-f001:**
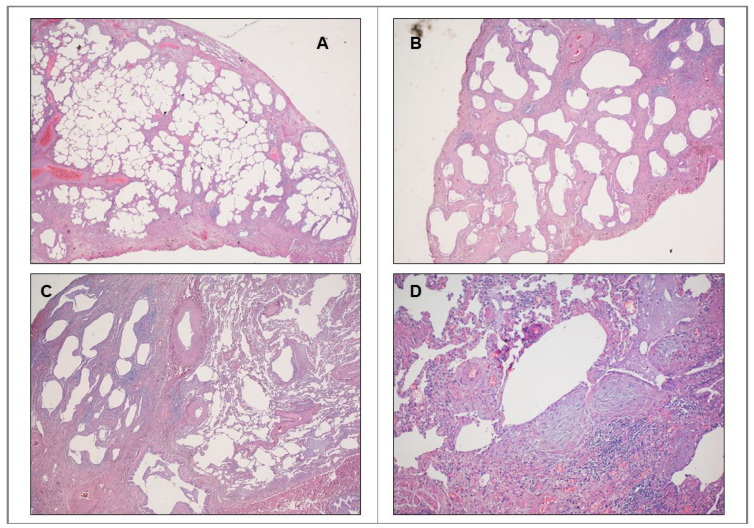
Histopathologic patterns of usual interstitial pneumonias. (**A**) Evidence of marked fibrosis/architectural distortion in a predominantly subpleural/paraseptal distribution. (**B**) Evidence of microscopic honeycombing. (**C**) Presence of patchy involvement of lung parenchyma by fibrosis (spatial heterogeneity). (**D**) Presence of fibroblast foci (temporal heterogeneity). Their detection allows UIP diagnosis confirmation.

**Figure 2 ijms-22-02882-f002:**
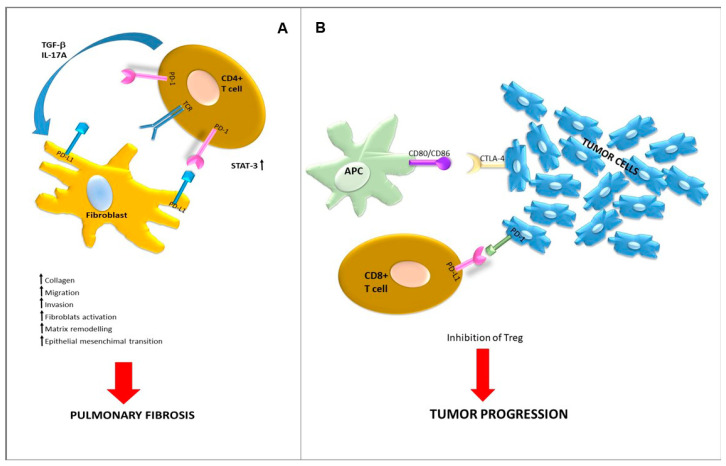
(**A**) Role of PD-1/PDL11 axis in development of pulmonary fibrosis. PD1 is expressed on CD4+ T cells whereas PD-L1 is expressed on fibroblasts. The interaction between receptor and ligand promotes IL17A and TGF-β production by the CD4+ T cells, through STAT 3, thus inducing pro-fibrotic responses by fibroblasts. (**B**) Role of PD-1/PDL-1 axis in the immune escape leading to progression of NSCLC. PD-1 is expressed by activated cytotoxic T cells whereas PDL-1 is expressed by lung cancer cells. The subsequent interconnection allows malignant cells to avoid the immune response, leading to tumor progression.

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
