# Peer review of "Immune Stroma in Lung Cancer and Idiopathic Pulmonary Fibrosis: A Common Biologic Landscape?"

_ijms, 2021, doi:10.3390/ijms22062882_

Round 1
Reviewer 1 Report
This review, although interesting and full of ideas, should be better structured in some parts, which are difficult to understand at first reading. Attention to style and a few errors that could have been avoided.
A few examples:
-attention to biobliography (I don't think this is MDPI's style) and the use of acronyms in the text;
-Figure 1 is never mentioned in the text. Instead, Figure 2 should be placed immediately after its citation in the text;
-line 64: change animal models to in vivo;
-line 99: change TRKs to RTKs;
-paragraph 2.2: changing line spacing in the text;
-line 109: change mTor to mTOR and report always the same wording;
-line 205: add spaces;
-line 209: change represent to represents;
-line 264: change remain to remains;
-line 317: EMT has been previously described by its acronym;
-line334: IC to ICs
Attention in the final bibliography the text is underlined in green
Author Response
Reply to Reviewer 1
Comments and Suggestions for Authors
This review, although interesting and full of ideas, should be better structured in some parts, which are difficult to understand at first reading. Attention to style and a few errors that could have been avoided.
We really thank the Reviewer for comments and for suggestions which improve the quality of the manuscript. We have revised and restructured the manuscript. Below the point-by-point answers (A).
A few examples:
-attention to biobliography (I don't think this is MDPI's style) and the use of acronyms in the text;
-Figure 1 is never mentioned in the text. Instead, Figure 2 should be placed immediately after its citation in the text;
-line 64: change animal models to in vivo;
-line 99: change TRKs to RTKs;
-paragraph 2.2: changing line spacing in the text;
-line 109: change mTor to mTOR and report always the same wording;
-line 205: add spaces;
-line 209: change represent to represents;
-line 264: change remain to remains;
-line 317: EMT has been previously described by its acronym;
-line334: IC to ICs
Attention in the final bibliography the text is underlined in green
- We really thank the Reviewer for careful revision of the text. Reference style has been revised, figure quotation and position have been modified, typing errors have been revised as suggested.
Reviewer 2 Report
Comments:
Please standardize number of citations.
1- Introduction (page 1, line 33) you reported the bad prgnosis associated with IPF. However, this data referred to the pre-antifibrotic era in the management of IPF. Pirfenidone and nintedanib treatment has significantly improved survival and reduced disease progression, although no significant effect has never been demonstrated in terms of quality of life (https://pubmed.ncbi.nlm.nih.gov/33102528/). Please, update this section.
2- You should add a specific paragraph regarding the available biomarkers shared by IPF and lung cancer, as described by the following references:
https://pubmed.ncbi.nlm.nih.gov/29150411/
https://pubmed.ncbi.nlm.nih.gov/33572642/
https://pubmed.ncbi.nlm.nih.gov/32940077/
https://pubmed.ncbi.nlm.nih.gov/32969271/
3- significantly, one of the drugs currently approved for the treatment of IPF (nintedanib) has been previously used as second-line chemotherapy for lung cancer: you should analyze this aspect in your review (https://pubmed.ncbi.nlm.nih.gov/32896631/; https://pubmed.ncbi.nlm.nih.gov/32748739/)
Author Response
Reply to Reviewer 2
Comments and Suggestions for Authors
Please standardize number of citations.
We really thank the Reviewer for careful reading of the manuscript and the constructive remarks. We have deeply revised the structure of the paper and the reference section. Below the point-by-point answers (A).
- Introduction (page 1, line 33) you reported the bad prgnosis associated with IPF. However, this data referred to the pre-antifibrotic era in the management of IPF. Pirfenidone and nintedanib treatment has significantly improved survival and reduced disease progression, although no significant effect has never been demonstrated in terms of quality of life (https://pubmed.ncbi.nlm.nih.gov/33102528/). Please, update this section.
A1. We thank the Reviewer for this fruitful comment, and we have modified the manuscript accordingly. The introduction has been implemented as follows: “It should be underlined that treatment with pirfenidone and nintedanib, currently approved in clinical use, has significantly improved survival and reduced disease progression, although no significant effect has never been demonstrated in quality of life”.
2- You should add a specific paragraph regarding the available biomarkers shared by IPF and lung cancer, as described by the following references:
https://pubmed.ncbi.nlm.nih.gov/29150411/
https://pubmed.ncbi.nlm.nih.gov/33572642/
https://pubmed.ncbi.nlm.nih.gov/32940077/
https://pubmed.ncbi.nlm.nih.gov/32969271/
A2. We thank the Reviewer for this fruitful comment. The text has been implemented as follows: ” Common available biomarkers. The biologic relation which subsists between cancer and IPF gives a strong rationale to the identification of available biomarker signatures able to stratify patients based on the risk of disease progression and poor outcome. The perspective observational PROFILE study allowed the generation of a panel of serum proteins - secreted from metaplastic epithelium, aberrantly activated in fibrogenic process - that most associated with IPF progression if compared to controls. They were surfactant protein D (SP-D), α2 macroglobulin, matrix metalloproteinase 7 (MMP7), T-cadherin, AXL receptor tyrosine-kinase, and MMP9 . Notably, high levels of the CA19-9 protein were significantly associated to progressive disease whereas high level of other proteins such as CA-125, macrophage migration inhibitory factor (MIF), carcinoembryonic antigen (CEA), free prostate-specific antigen (PSAf), and MMP7 were associated with a significant increase in overall mortality. Among those serum proteins, oncomarkers are routinely measured in biochemistry laboratories. Within respect to their predictive role, it has been reported that their combination in a specific signature (high levels of circulating CEA, Ca15.3, Ca19.9, Ca125, and KL-6) is clearly associated to IPF. Importantly, concomitant histologic studies, demonstrated that the Ca19.9 and Ca125 markers derived from epithelial damage. By stratifying patient on disease progression rate, Ca 19.9 level emerges as inversely correlated to lung function decline, mainly in those patients who rapidly progress. Increase Ca 19.9 levels are associated with increased levels in the end-stage IPF, irrespective on treatment setting. Indeed, fewer data are available on the potential role of circulating oncomarker levels in predicting response to therapy. The KL-6 levels have been reported to correlate with response to nintedanib response in IPF patients, since they decrease during treatment and in case of as KL-6 levels declined during treatment and were associated with pulmonary function tests performance. Similarly, changes in CA 15-3 levels predict variation of functional tests in IPF on antifibrotic treatment and coherently, higher levels are associated to worse outcome. This observation points out the role of Ca 15.3 marker as powerful and easily measurable follow up biomarker in IPF. Overall, the described serum biochemical signatures support the evidence of a common biologic landscape between IPF and (lung) cancer on one hand, whereas on the other represent easy, non-invasive and non-expensive reliable IPF markers, potentially powerful to predict response to drugs and disease progression (eventually to rapidly identify those patients who will deserve strict follow up and will be earlier addressed to lung transplantation).
- significantly, one of the drugs currently approved for the treatment of IPF (nintedanib) has been previously used as second-line chemotherapy for lung cancer: you should analyse this aspect in your review (https://pubmed.ncbi.nlm.nih.gov/32896631/; https://pubmed.ncbi.nlm.nih.gov/32748739/)
A3. We thank the Reviewer for this fruitful comment. The text has been implemented as follows - (line 232):” It should be underlined that nintedanib has been previously used as second-line therapy in combination with Docetaxel for advanced lung cancer. Recent bioinformatic analysis evaluating oncogenic TK expression in IPF, pointed out five microRNAs which act by modulating VEGF-A signalling pathway and epithelial to mesenchymal transition mechanisms in IPF. This evidence strongly supports the rationale of clinical use of nintedanib based on a cancer-related milieu which plays a central role in IPF”
Round 2
Reviewer 1 Report
The authors have improved the manuscript sufficiently